# Study of Ubiquitin Pathway Genes in a French Population with Amyotrophic Lateral Sclerosis: Focus on *HECW1* Encoding the E3 Ligase NEDL1

**DOI:** 10.3390/ijms24021268

**Published:** 2023-01-09

**Authors:** Shanez Haouari, Christian Robert Andres, Debora Lanznaster, Sylviane Marouillat, Céline Brulard, Audrey Dangoumau, Devina Ung, Charlotte Veyrat-Durebex, Frédéric Laumonnier, Hélène Blasco, Philippe Couratier, Philippe Corcia, Patrick Vourc’h

**Affiliations:** 1UMR 1253 iBrain, Université de Tours, Inserm, 10 Boulevard Tonnellé, 37032 Tours, France; 2Service de Biochimie et Biologie Moléculaire, CHU de Tours, 2 Boulevard Tonnellé, 37044 Tours, France; 3Centre SLA, CHU Limoges, 2 Avenue Martin Luther King, 87000 Limoges, France; 4Centre SLA, CHU Tours, 2 Boulevard Tonnellé, 37044 Tours, France

**Keywords:** amyotrophic lateral sclerosis, genetic, ALS, ubiquitin, *HECW1*

## Abstract

The ubiquitin pathway, one of the main actors regulating cell signaling processes and cellular protein homeostasis, is directly involved in the pathophysiology of amyotrophic lateral sclerosis (ALS). We first analyzed, by a next-generation sequencing (NGS) strategy, a series of genes of the ubiquitin pathway in two cohorts of familial and sporadic ALS patients comprising 176 ALS patients. We identified several pathogenic variants in different genes of this ubiquitin pathway already described in ALS, such as *FUS*, *CCNF* and *UBQLN2*. Other variants of interest were discovered in new genes studied in this disease, in particular in the *HECW1* gene. We have shown that the HECT E3 ligase called NEDL1, encoded by the *HECW1* gene, is expressed in neurons, mainly in their somas. Its overexpression is associated with increased cell death in vitro and, very interestingly, with the cytoplasmic mislocalization of TDP-43, a major protein involved in ALS. These results give new support for the role of the ubiquitin pathway in ALS, and suggest further studies of the *HECW1* gene and its protein NEDL1 in the pathophysiology of ALS.

## 1. Introduction

Amyotrophic lateral sclerosis (ALS) is the most frequent motor neuron disease in the adult. It is a fatal neurodegenerative disease characterized by the progressive death of cortical and spinal motor neurons. Patients suffering from ALS present progressive muscle atrophy and weakness. Death occurs 3 to 5 years after diagnosis by respiratory failure due mainly to impairment of respiratory function [1].

A hallmark of ALS is the accumulation of misfolded proteins and protein aggregates in degenerating motor neurons and astrocytes [2,3] This is primarily due to defects in proteostasis. Defects in the function or regulation of proteostasis have therefore been suspected in ALS for many years. A large proportion of the protein aggregates described in ALS patients are ubiquitin-positive, which has led to the suggestion of a role for the ubiquitin pathway in the etiology of ALS. Several recent observations in patients and in disease models provide further evidence for an important role of the ubiquitin pathway in the pathophysiology of ALS. For example, several genes directly involved in the ubiquitin pathway have been found to be mutated in ALS patients, including *UBQLN2* and *CCNF* [4,5]. Mutations of these genes results in impaired proteostasis and protein aggregation. Thus, processes participating in protein degradation, and in particular the ubiquitin pathway, play major roles in the pathophysiology of ALS [6].

The ubiquitin and ubiquitin-like (SUMO) pathways consist of intracellular pathways comprising three classes of enzymes, activating E1, and conjugating E2 and E3 ligases encoded by more than 700 genes in the human genome. E3 enzymes provide the specificity of the target proteins. They are involved in many cellular processes by regulating the function or half-life of target proteins via their post-translational modification, ubiquitinylation or SUMOylation [7]. These E3 enzymes are classified into two major families, the E3 RING ligases and the E3 HECT ligases. Each of these families contains enzymes playing important roles in mechanisms involved in neurodevelopment and neurodegeneration, such as the NEDD4 subfamily of HECT E3 ligases, several members of which have been implicated in the pathophysiology of neurodegenerative diseases [8,9]. The ubiquitin pathway also involves other proteins required for its function. One example is the valosin-containing protein (VCP, p97), which binds to ubiquitinated proteins and induces their degradation by the proteasome. Mutations in the *VCP* gene are found in about 2% of familial ALS cases [10].

Functional alterations of the ubiquitin system are thus one of the mechanisms involved in the pathophysiology of ALS. We first present here the results of a genetic study based on a targeted analysis of ubiquitin pathway genes in patients with familial or sporadic ALS. We then focused our study on the *HECW1* gene encoding the NEDL1 protein, a candidate in the pathophysiology of ALS as suggested by transgenic mice overexpressing human NEDL1 that exhibit muscle atrophy and motor neuron death [11].

## 2. Results

### 2.1. Genetic Screening of 12 Genes of the Ubiquitin Pathway in ALS

We selected 12 genes of the ubiquitin pathway, based on the fact that they have been already identified as involved in ALS pathology or they have been proposed as candidates in this disease. These genes were selected on the basis of knowledge available in ALS databases (ALSOD and MinE databases) for *VCP* and the E3 *FUS*, functional studies from our laboratory for the E2 *UBE2D2*, E3 *RBX1* and *UHRF2*, and data from the literature for E2 *UBE2D3*, as well as for E3 or members of E3 complexes such as *FBXO32, HECW1, KDM2B, MARCH5, RNF19A* and *TIM63* [12].

We analyzed these genes by targeted NGS sequencing using a Haloplex capture on French familial ALS (FALS) patients from 58 different families. The sex ratio (men/women) was 0.72 (27/37). Patients from these families did not carry an hexanucleotide repeat expansion (HRE) in intron 1 of the C9orf72 gene, the most frequent mutation in ALS, nor did they carry pathogenic or probably pathogenic variants in other 29 causative genes in ALS (listed in Section 4). We studied the exons and exon-intron junction regions of the 12 selected genes of the ubiquitin pathway, and considered only variants with a very low minor allele frequency (MAF) in the general population (GnomAD). Among them, we identified the presence of genetic variants meeting these criteria in five genes (Table 1). These variants were confirmed by Sanger sequencing. Three different variants were identified in FUS gene in three patients (5.1%), one pathogenic variant p.Arg521Cys (reported in ClinVar, class 5 according to the ACMG) and two probably pathogenic variants p.His517Tyr and p.Arg521Ser (class 4). The p.Arg521Cys variant was discovered in a female patient with an age of onset of 32 years old and a familial form of ALS. The new p.Arg521Ser variant was observed in a male patient with age of onset of 22 years old and a familial form of ALS, and the p.His517Tyr variant was discovered in a 77 year-old woman with a slowly progressing and sporadic pathology. Two variants were present in the KDM2B gene (1.2% patients), and two variants were present in the RNF19A and TRIM63 genes. Seven variants were observed in HECW1 gene in seven different patients (12.1%).

### 2.2. Genetic Screening on a Second Cohort of ALS Patients

In order to validate our result, we extended our study to a second cohort of 118 ALS patients. We also excluded from the study all patients carrying an HRE expansion in C9orf72 gene or a pathogenic or probably pathogenic variant in one of the 29 ALS causative genes (listed in Material and Methods section). DNA was analyzed using a new targeted capture (Twist) based on the analysis of four genes of the ubiquitin pathway, i.e., FUS and HECW1 genes, in which we found several variants in the first cohort of patients, and two new genes that were absent in the first design (first cohort), the CCNF and UBQLN2 genes. CCNF was studied because it was found to be mutated in familial ALS cases and it encodes cyclin F, a component of an E3 complex SCF (Cyclin F) [5]. The UBQLN2 gene is located on the X chromosome and encodes a shuttle protein interacting with both polyubiquitinated proteins and the proteasome [13]. UBQLN2 mutations were also identified in ALS patients [4].

The second cohort consisted of 90 patients with sporadic ALS and 37 patients with familial ALS. The sex ratio (men/women) in this second cohort was 1.3 (72/55). The genetic variants identified are described in Table 1. The ACMG classification was also used to propose a relative contribution of each of these variants in the pathogenesis of ALS. The novel and possibly pathogenic p.His517Tyr variant (class 4) was also identified in the FUS gene. Six variants were found in the CCNF gene, including two variants of unknown significance (VUS, class 3) and the p.Ser621Gly variant, which was reported as pathogenic in the first publication on CCNF involvement in ALS [5]. This pathogenic variant p.Ser621Gly was observed in a 74 year-old woman who presented a sporadic form of ALS, which initially appeared in the lower li*mbs.* The UBQLN2 gene showed only one VUS (class 3). The study of HECW1 in this second cohort revealed two variants already listed in GnomAD, as well as two new variants.

The *HECW1* gene encodes a 1606 amino acid residue protein, NEDL1 (NEDD4-like ubiquitin protein ligase-1), a member of the HECT-E3 ubiquitin ligase family. None of the 11 variants identified in the HECW1 gene in the first and second cohorts of patients were listed in ClinVar. Among them, we identified three heterozygous variants of particular interest because they were absent from the databases and were of unknown significance (Figure 1A). All of them were predicted to be pathogenic by the prediction software polyphen II (p.Asp598Tyr, score 0.997; p.Gly1246Ala, score 0.999; p.Arg1442Cys, score 1.000). The Asp598Tyr variant is located in the NEDL1 protein between the C2 domain in the N-terminal region and the first WW-rich domain. The Gly1246Ala variant is located just before the HECT domain, and the Arg1442Cys variant is in the middle of the HECT functional domain. The latter p.Arg144Cys variant in HECW1 was discovered in a 76 year-old patient in a family with ALS/FTD (Figure 1B). This position was highly conserved throughout evolution as shown by the comparison of NEDL1 sequences extracted from UniprotKB and aligned by MUSCLE (Figure 1C). To better understand the environment of this arginine 1442 in NEDL1, we interrogated the EMBL program AlphaFold that predicts the 3D structure of proteins (Figure 1D). Arginine 1442 is located in an alpha helix; it interacts with methionine 1443 in the same helix and with asparagine 1384 located in another alpha helix nearby. Thus, p.Arg1442Cys appears to be particularly involved in the three-dimensional structure of this catalytic region of NEDL1.

### 2.3. Study of Neural Expression of NEDL1 Encoded by HECW1

To better understand the function of the *HECW1* gene and the NEDL1 protein in the central nervous system, we performed in vivo and in vitro expression studies. The *Hecw1* gene was expressed in all regions of the central nervous system studied in mice, but not in the gastrocnemius muscle (Figure 2A). To go into more detail about the neuronal expression, we used developing primary hippocampal neuronal cultures from mouse embryos. The mouse *Hecw1* gene was found to be expressed at 7 and 14 DIV (Days of culture In Vitro) (Figure 2A). Neurons are still immature at 7 days and considered mature (functional synapses) at day 14, and NEDL1 was expressed at both stages. In order to study the expression of NEDL1 protein in cells in vitro, we used a plasmid expressing the wild-type human form of the protein harboring a Myc-FLAG tag at its C-terminus end. In transfected mature neurons, NEDL1 was localized preferentially in the soma, but also in some dendrites where punctate signals were visible, compared to the expression of the control protein GFP, which showed a more diffuse expression and presented more neuronal extensions. Similar experiments in human HEK293T cells confirmed a diffuse expression of NEDL1 in the cytoplasm (Figure 3A,B). Study of the R1442C mutant of NEDL1 showed similar expression in the cytoplasm, suggesting that the variant does not affect the cellular localization of the protein (Figure 3A,B).

### 2.4. Functional Study of the NEDL1 Protein

To better understand the function of NEDL1, we analyzed HEK293T cells transfected with the human NEDL1-myc protein. Compared to control conditions, we observed that NEDL1-expressing cells exhibited cell stress and appeared to enter a cell death process that we aimed to quantify. HEK293T cells, at 48 h post-transfection, were treated with propidium iodide and analyzed by flow cytometry (Figure 3C). Cells expressing human NEDL1 showed a two-fold increase in cell death (14.97% vs. 7.82% for control; *p* = 0.028). The human p.Arg1442Cys variant of NEDL1 also caused a significant increase in cell death, but was similar to that caused by the wild-type protein.

**Figure 3 ijms-24-01268-f003:**
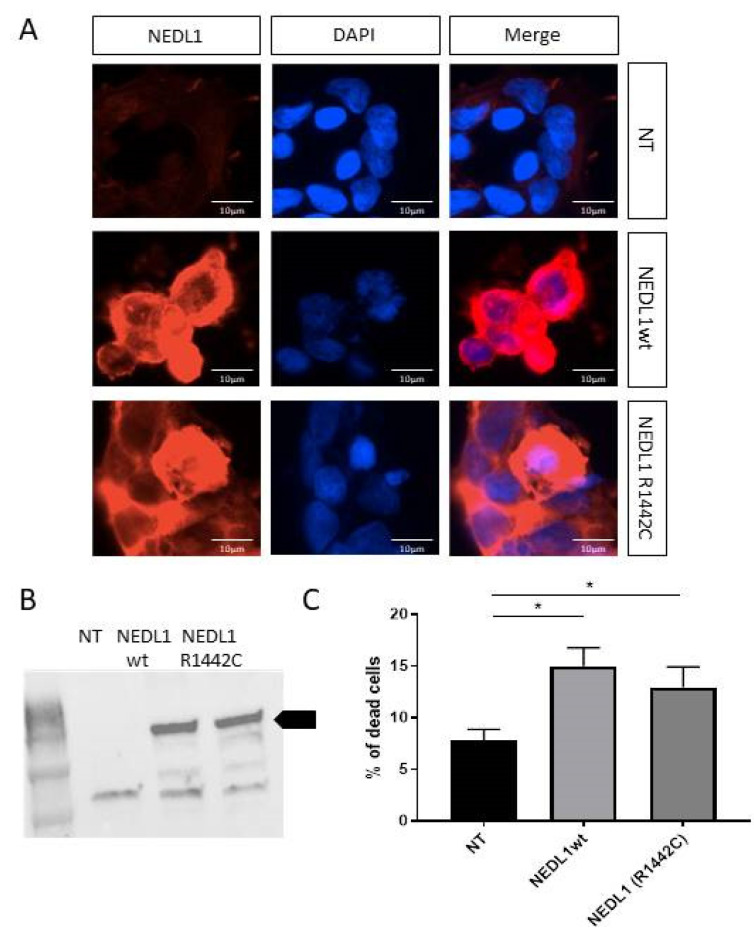
(**A**) NEDL1 protein expression (red) in untransfected HEK293T cells (NT, only transfection agent), and in HEK293 cells transfected with a plasmid expressing wild-type (WT) or mutated NEDL1 (p.Arg1442Cys) protein. The nucleus of the cells was stained with DAPI (blue). (**B**) Western blot confirmation of protein expression of wild-type and mutated forms of NEDL1 in HEK293T cells after transfection. The arrow indicates the band corresponding to NEDL1. (**C**) Viability assay showing a significant increase in cell death in HEK293T cells expressing wild-type or p.Arg1442Cys-mutated forms of NEDL1. * *p* < 0.05.

We then explored a possible link between NEDL1 and the pathophysiology of ALS. TDP-43 protein is a major component of ALS pathophysiology. Indeed, TDP-43 aggregates are found in the motor neurons of >95% of ALS patients. Overexpression of the human form of the E3 ligase NEDL1 in HEK293T cells did not show any change in TDP-43 protein concentration (Figure 4A,B). In contrast, immunocytochemistry analysis showed that overexpression of NEDL1 was associated with an increase in TDP-43 protein in the cytoplasm and a decrease in the nucleus, all without an overall decrease or increase in TDP-43 levels in the cells (Figure 4C). This observation is very interesting because the translocation of TDP-43 from the nucleus to the cytoplasm is a major feature of ALS. Overexpression of the p.Arg1442Cys variant of NEDL1 also results in mislocalization of TDP-43, but to a lesser extent than the wild-type protein (Figure 4C).

## 3. Discussion

The first objective of this study was to analyze a series of genes of the ubiquitin pathway in ALS in French patients. Particular attention was then paid to the *HECW1* gene and the protein it encodes, NEDL1, because several variants of interest were discovered in this gene during our study and a previous work had shown that its overexpression in mice led to motor neuron loss.

We therefore first identified pathogenic or probably pathogenic variants in ALS genes encoding proteins involved in the ubiquitin/ubiquitin-like pathways. The *FUS* gene is one of the major ALS genes. It is implicated in 3% of familial ALS cases end encodes the Fused in sarcoma (FUS) protein, an E3 SUMO ligase [16,17]. We identified three variants in this gene. The position p.His517Tyr is known to be mutated in several ALS cases in the literature as p.His517Gln. The fact that this position is mutated in ALS patients, that it is located in the nuclear localization signal (NLS) domain of FUS, and that a predictive bioinformatics tool describes it as pathogenic led us to classify it as probably pathogenic. We found two other variants in this same NLS C-terminus domain, at position 521: a new p.Arg521Ser variant and the p.Arg521Cys variant known to be pathogenic in ALS [18]. This Arg521Cys variant has been used to generate transgenic mouse models of the disease [19]. One of the latest genes identified in ALS is the *CCNF* gene, encoding cyclin F, a component of an E3 ubiquitin–protein ligase complex (SCF-Cyclin F) [20]. We identified several VUS of interest in *CCNF*, p.Arg521His and p.Pro549Ser, because of their localization in the protein, and one known pathogenic variant, p.Ser621Gly. It is the same variant as the first mutation identified in the *CCNF* gene in ALS [5].

The DNA sequencing study on two cohorts of ALS led to identify variants of interest in other genes involved in the ubiquitin pathway, and notably in the *HECW1* gene. We chose to include this gene in the NGS panel for several reasons. It encodes the HECT-type E3 NEDL1 (NEDD4-like ubiquitin protein Ligase-1) that specifically targets mutant forms of SOD1, one of the main ALS genes, to allow their degradation [21]. In addition, a study showed that transgenic mice overexpressing the human wild-type form of NEDL1 developed ALS-like symptoms, affecting motor neurons [11]. Here we report the first *HECW1* variants in an ALS population. Based on the limited data available on this *HECW1* gene and NEDL1 function, we had to place several variants in class 3 (variants of unknown significance). One of the variants, p.Arg1442Cys, was particularly interesting because it is located in the HECT catalytic domain of NEDL1. As we have observed, the Arginine 1442 is highly conserved and may have an important role in the three-dimensional structure of this catalytic HECT region that allows fixing the E2-bound ubiquitin and then transferring it to the target protein.

Functional studies are needed to better understand this *HECW1* gene and the functions of the E3 ligase NEDL1 in order to perhaps reclassify the previous variants. NEDL1 is a member of the NEDD4 subfamily that contains nine members, sharing common domains [9]. One target of NEDL1 ligase action could be the receptor tyrosine kinase ErbB4, which is mutated in some ALS cases [22,23]. Scribble, TTF-1, and Smad4 are other target proteins of this E3 ligase [24,25,26]. We first wanted to better understand the expression of NEDL1. We have shown that it is particularly expressed in the mouse central nervous system, and that it is expressed in both developing and mature neurons. This last observation is in relation with the fact that the NEDD4 family is known to be involved in the mechanisms of neurodevelopment and neurodegeneration [9]. NEDL1 cellular localization is cytoplasmic. In neurons, it seems to be particularly enriched in the soma, but it is also present in neurites. It would be interesting to investigate if NEDL1 could also be linked to the membrane, as it contains a C2 domain. This domain, in the NEDD4 protein, mediates calcium-dependent localization of the protein to the plasma membrane [27]. Studies should also be performed to determine whether NEDL1 is localized in dendritic spines at the synaptic level and whether its overexpression is associated with the formation of protein aggregates. We have indeed observed a punctiform expression of NEDL1 especially in neurites. Glial cells also play an important role in the pathophysiology of ALS. Zhang et al. [11] described, in mice overexpressing NEDL1, mild microglial activation in spinal cord tissue, with no change in microglial cell number. Further studies on a possible expression of NEDL1 in glial cells, microglia and astrocytes, and on the consequence of its action on motor neuron environment should be performed.

Our in vitro studies have shown that overexpression of NEDL1 leads to stress that appears to cause cell death. We have shown that this overexpression indeed reduces cell viability. A previous study in cancer highlighted that NEDL1 cooperates with p53to induce apoptosis [28]. This function of NEDL1 is independent of its HECT catalytic domain. This could explain why we did not observe any functional effect of the p.Arg1442Cys variant, compared to the wild-type. The transcription factor p53 appears to be very important in the pathophysiology of ALS. For example, knocking out p53 in the ALS mice model C9orf72 (the most frequently mutated gene in ALS) completely rescues neurons from degeneration [29].

An important link was also observed between NEDL1 and another major actor in ALS, the TDP-43 protein. The *TARDBP* gene encoding TDP-43 is mutated in 4–5% of familial ALS cases [16,30], and cytoplasmic aggregates of TDP-43 are found in the motor neurons in 95% of ALS patients. TDP-43, a multifunctional DNA/RNA binding protein, is normally mainly nuclear. Some E3 ligases have been shown to participate in the regulation of TDP-43 homeostasis. For example, the E3 Parkin ligase forms a multiprotein complex with TDP-43 that promotes the cytoplasmic accumulation of TDP-43 and its polyubiquitination [31]. Overexpression of another mutated protein of the ubiquitin pathway in ALS, ubiquilin-2, also results in cytoplasmic localization of TDP-43 [32]. Interestingly we have shown that overexpression of NEDL1 also changes the localization of TDP-43 in cells, with TDP-43 becoming predominantly cytoplasmic. In ALS, translocation of TDP-43 from the nucleus to the cytoplasm is a common feature of the disease The TDP-43 protein can be ubiquitinated, for example, at its NLS. Indeed, site-directed mutagenesis of lysine 84, located in the NLS, has shown its importance in the nuclear import of TDP-43 [33]. It would now be interesting to test whether NEDL1, a cytoplasmic E3 ligase, has the ability to ubiquitinate TDP-43, for example at lysine 84, and whether this alters the localization of TDP-43 in motor neurons. Studies have also reported the presence of TDP-43 aggregates also in glial cells, such as astrocytes and oligodendrocytes [34,35]. A role for NEDL1 on TDP-43 in glial cells should therefore be studied in order to search for a pathogenic non cell-autonomous dysfunction resulting in neuronal degeneration in ALS.

In summary, our results support the idea that the ubiquitin pathway plays an important role in the pathophysiology of ALS. They highlight a gene of interest in ALS and motor neuron diseases, *HECW1*, which encodes NEDL1, an E3 ligase that has been poorly studied to date in the central nervous system. NEDL1 is a member of the NEDD4 family of proteins, a family of particular interest in the context of neurodevelopmental and neurodegenerative diseases. Further work on this *HECW1* gene and NEDL1 protein is needed to better understand the links between this ubiquitin E3 ligase, the modification of the localization of TDP-43, and the neuronal death involved in ALS.

## 4. Materials and Methods

### 4.1. Patients with ALS

A total of 176 Caucasian ALS patients recruited within the French ALS Centers were included in the study. All cases fulfill the criteria of definite, probable laboratory-supported and probable ALS. All patients gave their informed consent for DNA analysis. Genomic DNA was extracted from blood (Qiasymphony, Qiagen). Patients were recruited in the Neurology departments of hospitals that were members of the French network FISLAN (network for rare diseases on amyotrophic lateral sclerosis). Genetic analyses were performed in the Department of Biochemistry and Molecular Biology of the University Hospital of Tours (Agence de la Biomedecine, approval of 8 October 2018), which is a reference medical biology laboratory for the genetics of ALS (NOR: SSAP2121661A). The first cohort of ALS patients was composed of 58 patients with familial history of ALS (FALS); the second cohort included 118 ALS patients. All patients were first studied for the *C9orf72* gene. Investigation of the GGGGCC (G4C2) hexanucleotide expansion in this *C9orf72* gene was performed by repeat-primed polymerase chain reaction (PCR) and gene scanning in a 3130xl Applied sequencer (Thermofisher) [30]. Data were analyzed using GeneScan software.

### 4.2. Genetic Study by Next Generation Sequencing

A first cohort of 58 ALS patients was analyzed using a HaloPlex target enrichment system (Agilent Technologies, Santa Clara, CA, USA). A second cohort of 118 ALS patients (FLS and SLAS) was analyzed using a Twist target enrichment system (Twist Bioscience, CA, USA). The coding regions and intron–exon boundaries of the genes were analyzed. The following genes, known to be involved in ALS, were present in both designs: *ALS2*, *ANG*, *CHCHD10*, *CHMP2B*, *DAO*, *DCTN1*, *DPYSL3*, *FIG4*, *FUS*, *GRN*, *MAPT*, *MATR3*, *NEFH*, *OPTN*, *PRPH*, *PSEN1*, *PSEN2*, *SETX*, *SIGMAR1*, *SOD1*, *SPG11*, *QSTM1*, *TAF15*, *TARDBP*, *TBK1*, *TREM2*, *TUBA4A*, *UBQLN2*, *VAPB* and *VCP.* The FUS gene, listed as a causative gene for ALS, encodes an E3 enzyme of the ubiquitin pathway. Other ubiquitin pathway genes that have been studied were the following: in the first design (cohort 1) they were *FBXO32*, *HECW1*, *MARCH5*, *RBX1*, *RNF19A*, *TRIM63*, *UBE2D2*, *UBED2D3*, *UHRF2;* in the second design (cohort 2) they were *HECW1*, *CCNF*, *UBQLN2*. Libraries were sequenced using a MiSeq sequencer (Illumina, San Diego, CA, USA) with 150 bp paired-end sequencing (MiSeqReagent Kit V2; 300cycles) The sequences were analyzed using the bioinformatics pipeline of the ALS Center of the Hospital of Tours and the human reference genome UCSC hg19. Reads were aligned with the BWA algorithm (v.0.7.17), and variants were named using GATK tools (v.3.4) and annotated using the ANNOVAR software. Coverage was analyzed with samtools (v.1.8); variants were selected with minimum 30x coverage. Allelic frequencies in Exact databases and 1000 Genomes Projects were below 0.01% for all populations. Variants identified by NGS were validated by Sanger sequencing using a 3130xl genetic Analyzer (ThermoFisher) and classified according to ACMG [36].

### 4.3. Plasmid Constructs

Wild-type NEDL1 cDNA was cloned into a pCMV6-Entry vector (ORIGENE, RC213181). We introduced the R1442C mutation into this plasmid by site-directed mutagenesis using the Q5^®^ Site-directed Mutagenesis kit (New England Biolabs, Ipswitch, MA, USA). Primers for mutagenesis were designed using NEBaseChanger. The plasmid sequences were then validated by Sanger sequencing.

### 4.4. Cell Cultures and Transfections

HEK293T cultures. Cells were used to perform a cell viability assay, RT-qPCR and immunocytochemistry. The cells were grown at 37 °C, 5% CO_2_ in DMEM 10% SVF. For the cell viability assay and RT-qPCR, 3 × 10^5^ cells/well were seeded in 6-well plates. For immunocytochemistry, 1.5 × 10^4^ cells/well were seeded in 24-well plates on glass coverslips coated with poly-D-lysine (Sigma-Aldrich, L’lsle-d’Abeau Chesnes, France).

Primary neuronal cultures. All mouse experiments were approved by the French Ministry of Research (project authorization number 01456.03). C57BL/6J mouse embryos (Janvier Labs, Le Genest-Saint-Isle, France) were used at embryonic day 17.5 and brain tissue was dissected on cold DPBS with 1% penicillin–streptomycin (PS) (Thermo Fisher Scientific, Illkirch-Graffenstaden, France). Then, hippocampi were incubated with papain (Worthington; 10 U/mL) for 22 min at 37 °C before mechanical dissociation in DMEM:F12 (Thermo Fisher Scientific, 31331093), 10% FBS (Eurobio, Les Ulis, France CVFSVF06-01) and centrifugation at 250 g for 5 min. Resuspension of the cell pellet was performed in Neurobasal Plus (Thermo Fisher Scientific, A3582901) supplemented with 1X B27 Plus (Thermo Fisher Scientific, A35828-01) and 0.5 mM L-Glutamine (Thermo Fisher Scientific, 25030149). Dissociated neurons were then seeded at 3 × 10^5^ cells per well in 6-well plates or 6 × 10^4^ cells per well in 24-well plates on glass coverslips coated with poly-D-lysine (Sigma-Aldrich, P7280) and laminin (4 µg/mL) (Thermo Fisher Scientific, 23017-015). Half of the medium was changed twice a week.

Transfection was performed 24 h after seeding for HEK293T and at DIV11 for hippocampal neurons using Lipofectamine2000 (Thermo Fisher Scientific) according to the manufacturer’s instructions. For 6-well plates, a total of 2.5 µg of wild-type or p.Arg1442Cys NEDL1 plasmids were used for transfection in each well for 48 h. For 24-well plates, a total of 500 ng of wild-type or p.Arg1442Cys NEDL1 plasmids were used for transfection for 48 h.

### 4.5. RNA Extraction

Total RNA was extracted from primary neuronal cultures at DIV 7 and 14, and from HEK293T cells. Total RNA was also extracted from nervous structures (motor cortex, striatum, cerebellum, spinal cord) and muscle (gastrocnemius muscle) dissected from adult C57BL/6J mice. RNA extractions were performed using Trizol™ reagent and commercial Direct-Zol™ RNA Miniprep Plus (Zymo Research, Irvine, CA, USA) according to the manufacturer’s instructions. RNA concentrations and purity were assessed with a NanoDrop™ 2000/2000c spectrophotometry (Thermo Fisher Scientific).

### 4.6. Gene Expression Studies

For RT-PCR, cDNAs were obtained using 200 ng of total RNA with the sensiFAST cDNA synthesis kit (Meridian Bioscience, Memphis, TN, USA). PCR was then performed with Q5^®^ High-Fidelity DNA Polymerase (New England BioLabs) according to the manufacturer’s protocol, and PCR products were observed on a TBE 1% agarose gel.

For RT-qPCR, total RNA extraction and cDNA synthesis were performed as described above. Real-time PCR was performed on 5 ng of cDNA using the SyBR green Takyon kit (Eurogentec, Seraing, Belgium UF-NSMT-B0701) in a Light Cycler 480 (Roche). Data were normalized to *GAPDH* and transfectant-only condition using the 2^−∆∆Cp^ method. Primers used for human *HECW1*: GTTTTGTGTCCTTGCCCACT, GAATTGCAGCTGTCCACTCA; mouse *Hecw1:* ACTCCATAATTCCCAGCCAAT, AGCCTCCCAGTTTGGTGGA; GAPDH: CTGCACCACCAACTGCTTAG, GTCTTCTGGGTGGCAGTGAT.

### 4.7. Cell Viability Assay

Cells were harvested with trypsin, counted with trypan blue, and centrifuged at 900× *g* for 5 min before resuspension in DPBS at a concentration of 1 × 10^6^ cells/mL. The cells were then incubated with propidium iodide from Sigma Aldrich (10 µg/mL) for 30 min at 37 °C in the dark. After centrifugation at 900× *g* for 5 min, the cells were resuspended in DPBS. Quantification of live cells was performed by flow cytometry on a Becton Dickinson Accuri™ C6 Plus flow cytometer.

### 4.8. Immunocytochemistry

Cells on glass coverslips were first fixed in a solution of 4% paraformaldehyde, 4% sucrose in PBS for 20 min at room temperature. After washes in DPBS, the fixed cells were incubated for 1 h at room temperature with a solution of DPBS with 10% donkey serum from Sigma-Aldrich and 0.2% Triton X-100 (Sigma-Aldrich). Next, cells were incubated with anti-TDP-43 (Proteintech 10782-2-AP) and anti-FLAG (Sigma Aldrich F3165) primary antibodies diluted 1:500 in DPBS, 3% donkey serum, and 0.2% Triton X-100 for 1 h at room temperature. After three 10 min washes of DPBS, cells were incubated with fluorophore-coupled secondary antibodies diluted 1:300 in DPBS (Thermo Fisher Scientific A11008 and A21203) for 1 h at room temperature. Glass coverslips were then mounted with DAPI ProLong™ Diamond antifade solution (Fischer Scientific) after three 10-min washes of DPBS.

### 4.9. Statistical Analysis

Data analysis was performed with GraphPad Prism 8, using the Mann–Whitney test from at least three independent experiments, and differences were considered significant with *p* < 0.05.

## Figures and Tables

**Figure 1 ijms-24-01268-f001:**
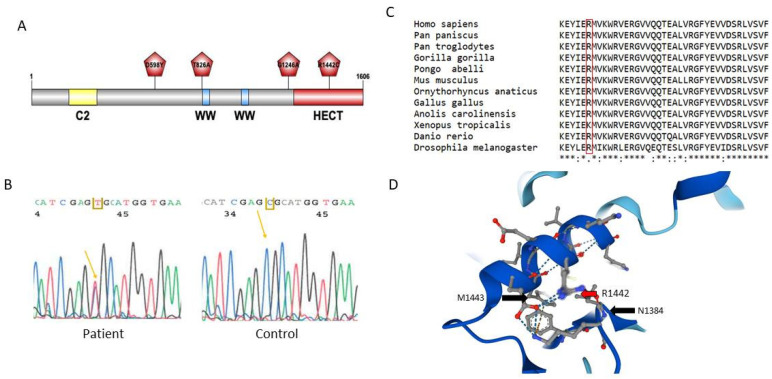
(**A**) Schematic representation of the NEDL1 protein. Functional domains consist of a C2 domain, two WW domains and a HECT catalytic domain. The position of variants of interest identified in ALS patients is indicated. (**B**) Electropherogram showing the heterozygous C>T variant in a patient responsible for the p.Arg1442Cys variation. (**C**) Alignment of the NEDL1 region embedding Arg1442 in different species. * same amino acid, . semi-conservative substitution of amino acids (amino acids with similar forms), : conservative substitution of amino acids. (**D**) Arg1442 amino acid region in the HECT domain of the NEDL1 protein; Alphafold Database; [14,15].

**Figure 2 ijms-24-01268-f002:**
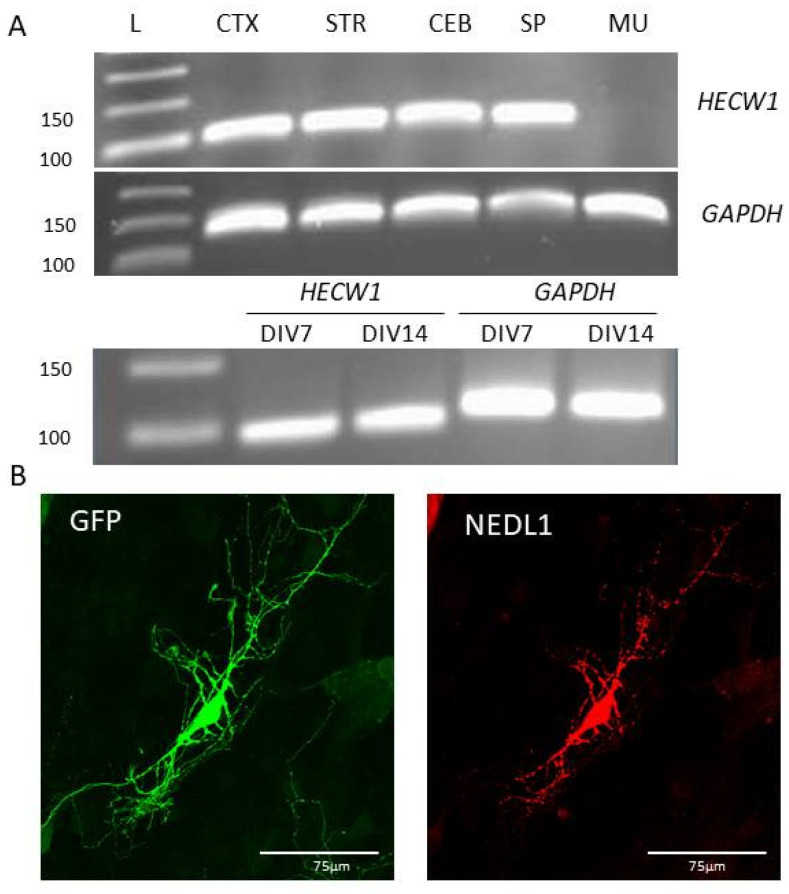
(**A**) Relative expression of the *HECW1* gene in different regions of the mouse central nervous system (CTX, Cortex; STR, Striatum; CEB, Cerebellum; SP: Spinal cord; MU, Muscle) (upper part of the figure), and in mouse hippocampal neuron cultures at days 7 and 14 in vitro (DIV) (lower part of the figure). (**B**) Immunocytochemistry on cultures of mature neurons showing localization of GFP (green) and NEDL1 (red) proteins. Scale bar: 75 μm.

**Figure 4 ijms-24-01268-f004:**
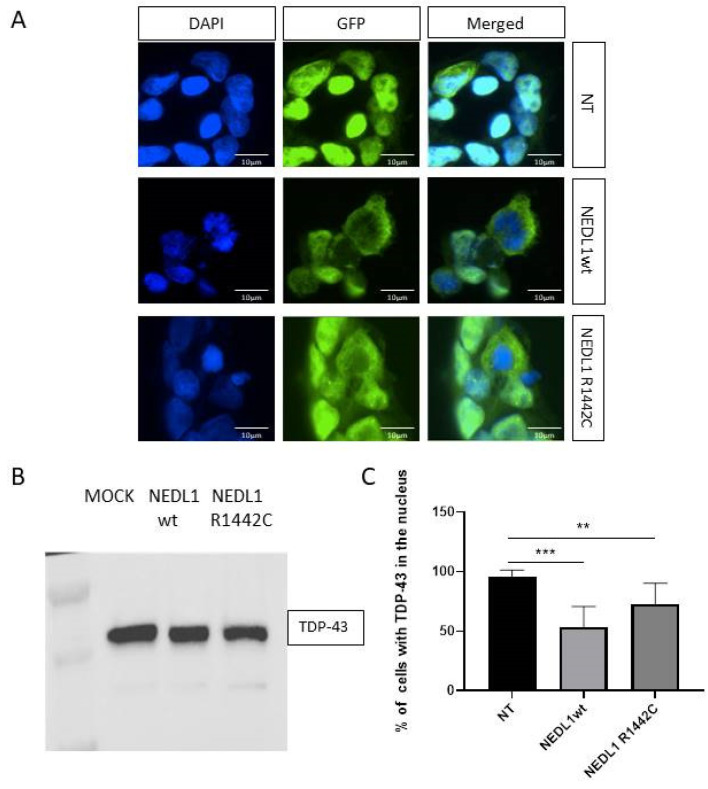
(**A**) TDP-43 protein expression (green) in untransfected HEK293T cells (NT, only transfection agent), and in HEK293T cells transfected with a plasmid expressing wild-type (WT) or mutated NEDL1 (p.Arg1442Cys) protein. Nuclei are stained with DAPI (blue). (**B**) Western blot showing expression of endogenous TDP-43 protein in all 3 conditions, untransfected HEK293T cells, or cells transfected with plasmids expressing NEDL1 WT or mutated NEDL1 (R1442C). (**C**) Percentage of untransfected or transfected HEK293T cells (WT or p.Arg1442Cys mutant) showing TDP-43 expression in the nucleus, normal localization of TDP-43. ** *p* < 0.05, *** *p* < 0.02.

**Table 1 ijms-24-01268-t001:** Variants observed in ALS patients.

Genes	Locations	Proteins	Functions	Variants in the Present Study	Cohort Number	Freq. MinE ALS Cases #	Freq. MinE Controls #	Freq. gnomAD	Condition in ClinVar	Class (ACMG)
*CCNF*	16p13.3	Cyclin F	Element of E3 ligase complex	S3G	2	0	0	0.00006	NR	2
				A74T	2	0.00015	0	0.00006	NR	2
				R406Q	2			0.00255	LB	2
				R521H	2	0	0	0.00001	NR	3
				P549S	2	0	0	0.00001	NR	3
				S621G	2	0	0	0.00042	P	5
*FBXO32*	8q24.13	F-box protein 32	E3 ligase	*	1					
*FUS*	16p11.2	FUS	E3 ligase, RBP	H517T	2	0	0	0	NR	4
				R518K	1	0	0	0	NR	3
				R521S	1	0	0	0	NR	4
				R521C	1	0	0	0.00001	P	5
*HECW1*	7p14.1-p13	NEDL1	E3 ligase	V196I	1	0	0.00546	0.00003	NR	3
				E502Q	1	0.00515	0.00300	0.00223	NR	2
				G576D	1	0.00046	0.00055	0.00061	NR	2
				D598Y	2	0	0	0	NR	3
				E820G	2	0	0	0.00268	NR	2
				T826A	1	0	0	0	NR	3
				D1005E	1	0.00241	0.00164	0.00184	NR	2
				V1099L	2	0.00252	0.00328	0.00061	NR	3
				V1184T	2	0	0	0	NR	2
				G1246A	1	0	0	0	NR	3
				R1442C	1	0	0	0	NR	3
*KDM2B*	12q24.31	Lysine demethylase 2B	Subunit CUL1-RING E3 ligase	H19R	1	0.00733	0.00655	0.00714	NR	2
				T28fs	1	0.00470	0.00300	0.00304	NR	2
*MARCH5*	10q23.32-33	RNF135	E3 ligase	*	1					
*RBX1*	22q13.2	Ring Box 1	E3 ligase	*	1					
*RNF19A*	8q22.2	Ring finger protein 19A	E3 ligase	A784T	1	0.00115	0	0.00007	NR	2
*TRIM63*	1p36.11	TRIM63	E3 ligase	A48V	1	0.00241	0.00300	0.00808	NR	2
*UBE2D2*	5q31.2	UBE2D2	E2 conjugating enzyme	*	1					
*UBE2D3*	4q24	UBE2D3	E2 conjugating enzyme	*	1					
*UBQLN2*	Xp11.21	Ubiquilin 2	Shuttle protein in the Ub system	S371N	2	0	0	0	NR	3
*UHRF2*	9p24.1	UHRF2	E3 ligase	*	1					
*VCP*	9p13.3	Valosin containing protein	Binds ubiquitynated substrates	*	1					

* no variant detected with a frequency below 0.01 in control databases. # ALS database MinE (www.projectmine.com accessed on 8 November 2022). NR: Not reported, P: Pathogenic, LB: Likely benin. Classication ACMG: 5 Pathogenic, 4 Likely pathogenic, 3 Variant of unknown significance, 2 Likely pathogenic.

## Data Availability

Not applicable.

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
