# Peer review of "Study of Ubiquitin Pathway Genes in a French Population with Amyotrophic Lateral Sclerosis: Focus on HECW1 Encoding the E3 Ligase NEDL1"

_ijms, 2023, doi:10.3390/ijms24021268_

Round 1

Reviewer 1 Report

The novelty of the manuscript is good .The quality of the manuscript is good and it does need extensive improvement before publication. The superiority of  ubiquitin pathway role in the pathophysiology of Amyotrophic Lateral Sclerosis be more emphasized.

The manuscript should be extended in scientific discussion. The authors presented their results and compared to some works, but did not present explanations for the reasons to reach these results.

Author Response

We thank the reviewer for this comment. We have added information in the introduction about the superiority of ubiquitin pathway role in the pathophysiology of ALS as requested: “Several recent observations in patients and in disease models provide further evidence for an important role of the ubiquitin pathway in the pathophysiology of ALS. For example, several genes directly involved in the ubiquitin pathway have been found to be mutated in ALS patients, including UBQLN2 and CCNF (Deng et al., 2011; Willians et al. 2016). Mutations of these genes results in impaired proteostasis and protein aggregation”.

Part of the manuscript consists of the results of the functional studies that we present in the Results section. We completed the discussion as requested by the reviewer to indicate why we reach these results: "Our in vitro studies have shown that overexpression of NEDL1 leads to stress that appears to cause cell death. We have shown that this overexpression indeed reduces cell viability. Previous study in cancer highlighted that NEDL1 cooperates with p53to induce apoptosis (Li et al., 2008).” In addition, we have separated the paragraph linking NEDL1 and TDP-43 to make it more visible in relation to the reviewer's point. We also modified in the discussion the sentence “An important link was also observed between NEDL1 and another major actor In ALS, the TDP-43 protein”. We also added in the discussion the sentence. “Some E3 ligases have been shown to participate in the regulation of TDP-43 homeostasis. For example, the E3 Parkin ligase forms a multiprotein complex with TDP-43 that promotes cytoplasmic accumulation of TDP-43 and its polyubiquitination (Hebron et al., 2013)”.

Reviewer 2 Report

The Authors, Drs Haouari et al., submitted an article in which report the identification of several pathogenic variants in different genes of the ubiquitin pathway already described in amyotrophic lateral sclerosis, such as FUS, CCNF and UBQLN2. Other variants of interest were discovered in new genes studied in this disease, in particular in the HECW1 gene.

The manuscript needs more clinical data of patients. One table showing detailed clinical features of affected subjects should be included in the text.

Author Response

We have added in the results section information available on patients with class 4 and 5 variants, or HECW1 variant analyzed in functional studies: “The p.Arg521Cys variant was discovered in a female patient with an age of onset of 32 years old and a familial form of ALS. The new p.Arg521Ser variant was observed in a male patient with age of onset of 22 years old and a familial form of ALS, and the p.His517Tyr variant was discovered in a 77-year-old woman with a slowly progressing and sporadic pathology.” “This pathogenic variant p.Ser621Gly was observed in a 74 year old woman who presented a sporadic form of ALS, which initially appeared in the lower limbs.” “The latter p.Arg144Cys variant in HECW1 was discovered in a 76-year-old patient in a family with ALS/FTD”.

Reviewer 3 Report

The manuscript "Study of ubiquitin pathway genes in a French population with 2 amyotrophic lateral sclerosis: focus on HECW1 encoding the E3 3 ligase NEDL1. " by Shanez Haouari is adequate for publication. The idea is ok, and this referee think is a good approach. But the manuscript needs major and minor revisions.

Major revision

1.- The author used primary hippocampal neuronal cultures from mouse embryos, but they did not use astrocytes in primary culture. Why did not try to use. Astrocytes still immature at day 14 and considered mature at day 21. They will indicate if NEDL1 is expressed at both stages in astrocytes. Authors can use GFAP (glial fibrillary acidic protein) to detect the presence of astrocytes and NEDL1 protein in astrocytes.

2.- Determination of the cell death in astrocytes and in mixer culture (astrocytes-neurons) will give information about the mechanisms. Cell expression of NEDL1 and cell death must be evaluated in astrocytes too.

3.- Inflammation is normally unit to oxidative stress in many neurodegenerative diseases, with a clear inflammation and probably oxidative stress. So can you determine some parameter of inflammation.

4.- TDP-43 protein is a major component of ALS pathophysiology. Indeed, TDP-43 aggregates are found in the motor neurons of >95% of ALS patients, but what happen with the other cells inside Central Nervous System (astrocytes, microglia, oligodendroglia). Oligodendroglia is a type of cells important in ALS.

5.- Authors indicated “translocation of TDP-43 from the nucleus to the cytoplasm is a major feature of ALS and the overexpression of the p. Arg1442Cys variant of NEDL1 showed the same effect on TDP-43 mislocalization in the cytoplasm”. Authors can not indicate this strong relationship because they are not enough results to indicate that.

Minor revision

1.- Sometimes there are space between letters, others no.

2.- References will be all at well disposition.

Author Response

1, 2) We focused our studies on NEDL1 on neurons, which degenerate in ALS. Indeed Zhang et al (2011) observed that overexpression of NEDL1 leads to motor neuron death in transgenic mice. We have the neuronal culture protocol in routine in our laboratory. For the functional studies we used a tagged NEDL1 protein, because there is no commercial antibody of good quality for NEDL1. The immunocytochemistry experiment was designed to study the subcellular localization of NEDL1 in neurons. We are not used to the primary glial cell culture protocol. As suggested by the reviewer, it would be interesting to study the expression and role of NEDL1 in glial cells and in particular microglial cells. Zhang et al, 2011, indeed showed that overexpression of NEDL1 in mice was associated with mild but significant microglial activation (this without change in microglial cell number). We have added this point in the discussion: « Glial cells also play an important role in the pathophysiology of ALS. Zhang et al (2011) described in mice overexpressing NEDL1 mild microglial activation in spinal cord tissue, with no change in microglial cell number. Further studies on a possible expression of NEDL1 in glial cells, microglia and astrocytes, and on the consequence of its action on motor neuron environment should be performed.”

3) An important question raised by the reviewer is the role of inflammation in ALS. ALS is characterized in vivo by a complex mechanism of macrophage and lymphocyte infiltration, activation of astrocytes and microglia, among others. It would be interesting to study if NEDL1 plays a role in these mechanisms, this could be an important study that we will certainly carry out in our laboratory in the near future using different cell models that we need to set up. The sentence we added in the discussion section (noted in the response to point 1-2 above) indicates this idea.

4) The presence of TDP-43 aggregates in degenerating motor neurons is a hallmark of ALS. Understanding the role of these aggregates located in motor neurons in the death of these motor neurons is a crucial question. As indicated by the reviewer, TDP-43 positive aggregates are also present in glial cells. Future studies should also focus on these issues and then also study NEDL1 in this context. This important point raised by the reviewer was added in the Discussion section: we removed the sentence “Abnormalities in wild-type (overexpression) or mutant TDP-43 is known to induce p53-dependent cell death in neural cell lines (Vogt et al. 2018) », and replaced it by “Studies have also reported the presence of TDP-43 aggregates in glial cells, such as astrocytes and oligodendrocytes (Arai et al., 2010; Prater et al., 2021). A role for NEDL1 on TDP-43 in glial cells should therefore be studied in order to search for a pathogenic non cell-autonomous dysfunction resulting in neuronal degeneration in ALS.”

5) We have included the reviewer's comment on the mutated form of NEDL1 by changing a sentence in the result section: “Overexpression of the p.Arg1442Cys variant of NEDL1 also results in mislocalization of TDP-43, but to a lesser extent than the wild type protein (Figure 4C).”

Minor revisions:

We removed the spaces between the letters when we saw them.

We have placed the references in the right positions.

Round 2

Reviewer 3 Report

accepted